# Clinical Features and Risk Factors of Postoperative Stroke in Adult Moyamoya Disease

**DOI:** 10.3390/brainsci13121696

**Published:** 2023-12-08

**Authors:** Wen Liu, Kaixin Huang, Jianjian Zhang, Da Zhou, Jincao Chen

**Affiliations:** 1Department of Neurosurgery, Zhongnan Hospital, Wuhan University, Donghu Road 169, Wuhan 430071, China; liuwen9207@163.com (W.L.); whuhuangkx@163.com (K.H.); 18627121530@163.com (J.Z.); 2Neuroepigenetic Research Lab, Medical Research Institute, Wuhan University, Donghu Road 115, Wuhan 430071, China; 3Center for Health Information and Statistics of Hubei, Wuhan 430071, China; zhangstereo123@126.com

**Keywords:** Moyamoya disease, adult, revascularization, postoperative hemorrhagic stroke, postoperative ischemic stroke, risk factors

## Abstract

Background and purpose: The clinical features of and risk factors for postoperative stroke after surgical revascularization in adult moyamoya disease (MMD) have not been fully elucidated. To this end, the baseline clinical features were hereby described, and the risk factors for postoperative stroke were determined. Methods: Data of 4078 MMD inpatients were collected retrospectively across all secondary- and higher-level hospitals of Hubei Province from January 2019 to December 2020. In accordance with inclusion and exclusion criteria, 559 adult MMD inpatients were finally enrolled. The associated characteristics and potential risk factors were analyzed, and the Kaplan–Meier risk of stroke was also calculated. Results: The patients consisted of 286 females and 273 males, with a mean age of 49.1 ± 10.0 years, all of whom had at least 1 year of follow-up (median 25.1 months). There were 356 cases of preoperative ischemic symptoms and 203 cases of preoperative hemorrhage symptoms. Indirect, direct, and combined revascularization were conducted on 97, 105 and 357 patients, respectively. Among these patients, 17 had postoperative hemorrhagic stroke (PHS), and 43 had postoperative ischemic stroke (PIS). A comparison between PHS/PIS group and control group (patients without postoperative stroke events) showed that preoperative hemorrhage was significantly associated with PHS (*p* = 0.003), while hypertension (*p* = 0.003), diabetes mellitus (*p* = 0.003) and modified Rankin scale (mRS) (*p* = 0.034) at admission were associated with a higher rate of PIS. Furthermore, preoperative hemorrhagic stroke was identified as a risk factor for PHS (odds ratio [OR], 4.229 [95% CI, 1.244–14.376]; *p* = 0.021), while hypertension (odds ratio [OR], 0.424 [95% CI, 0.210–0.855]; *p* = 0.017), diabetes mellitus (odds ratio [OR], 0.368 [95% CI, 0.163–0.827]; *p* = 0.016) and admission mRS (odds ratio [OR], 2.301 [95% CI, 1.157–4.575]; *p* = 0.017) were found to be risk factors for PIS. Conclusions: The age distribution of adult MMD patients with revascularization was predominantly concentrated within the range from 46 to 55 years. Preoperative hemorrhage events were considered the risk factor for PHS. Hypertension, diabetes and admission mRS were correlated with PIS, and were also the risk factors for PIS. These results indicated the possible contribution of enhancing systematic disease management to the prevention of postoperative cerebrovascular accidents.

## 1. Introduction

Moyamoya disease (MMD) is a rare pathological cerebrovascular disorder characterized by progressive stenosis of the intracranial internal carotid arteries and proximal branches, resulting in the formation of compensatory tiny vessels resembling “puffs of smoke” [1]. This rare cerebrovascular disorder exhibits significant regional and ethnic differences, and occurs more frequently in China, South Korea, and Japan than in other countries [2]. In the United States, however, an upward trend has been observed [3]. Although the molecular mechanism of MMD has not been fully elucidated, several gene variants and monogenic diseases have been linked to moyamoya angiogenesis [4], among which RNF213 has been proposed as a major founder variant in MMD patients from east Asia [5].

Despite advances in understanding the pathophysiological and genetic pathophysiological basis of MMD, disease-specific drug treatment has not yet been reported [6]. Currently, surgical revascularization is recommended for symptomatic MMD patients. Three major forms of revascularization, i.e., direct, combined and indirect, are generally used [7]. Several studies reported that all three revascularization procedures in adult patients could prevent stroke events [8,9,10,11]. However, the incidences of postoperative ischemic/hemorrhagic stroke still reached 5.4–5.5% [12]. To date, general risk factors for stroke in adults with MMD who have had surgical revascularization have not been sufficiently investigated [13,14].

To this end, the present study was carried out to identify clinical features and general risk factors of postoperative hemorrhagic or ischemic cerebrovascular events in adult MMD patients. In this case-control retrospectively study of adult MMD patients, many background factors and different surgical procedures were analyzed.

## 2. Materials and Methods

### 2.1. Ethics

Inpatient information was collected from the inpatient electronic system database of Hubei Province, which is operated by the provincial health committee of the Hubei Disease Management and Data Center. To preserve patient privacy, social security card number, ID card number, contact information, and personal address were anonymized in compliance with the statistical legislation and ethical requirements of the People’s Republic of China. This study was approved by the Ethics Committee of Zhongnan Hospital of Wuhan University (approved number: 2023025K, approved date: 11 April 2023). 

### 2.2. Data Description and Collection

MMD frequently occurs in Hubei Province, an area located in central China with a population of 58.85 million (standardized incidence rate > 1.5 per 100,000 inhabitants per year) [15]. To promote the standardization and uniformity of healthcare services in hospitals, as well as to strengthen the supervision of service quality, the National Health Commission of China (NHC) has proposed the extensive implementation of electronic medical records (EMRs) in healthcare institutions. This proactive measure is designed to guarantee the scientific accuracy of health service databases and streamline the process of evidence-based supervision and evaluation of medical quality. Since 2016, EMRs have been implemented in Hubei Province to improve the standardization and homogeneity of healthcare capacity, which ensures the accuracy of inpatient data while providing evidence-based information. This initiative aims to enhance statistical analysis capabilities for medical services provided to inpatients in 929 public hospitals, all of which are classified at or above the second-tier level, across thirteen cities. Hospital grades, used as an evaluative benchmark, consider several factors, including hospital size, focus on scientific research, staff and technical expertise, medical facilities, and more. Based on a comprehensive evaluation considering these aspects, hospitals are assessed and categorized into one of three levels, with a higher level indicating more advanced qualifications. Specifically, second-tier hospitals, functioning as regional healthcare centers, offer services to multiple communities. They act as technical hubs for regional medical prevention, focusing on monitoring high-risk groups, handling first-level referrals, offering operational and technical support to first-level hospitals, and contributing to teaching and scientific research. In this research, EMRs of MMD inpatients of Hubei at secondary- and higher-level hospitals were retrieved, which, taking into account the availability and quality of data, was operated by the provincial health committee of the Hubei Disease Management and Data Center. The collected data included basic demographic information, detailed disease diagnoses, and surgical procedures outlined by the International Classification of Diseases (ICD-10).

### 2.3. Study Population

Patients with a clinical diagnosis of MMD were identified from the database of EMRs from 1 January 2019, to 31 December 2020. The diagnosis of MMD was made based on clinically accepted guidelines: (1) stenosis or occlusion at the carotid artery’s end, proximal ACA, and/or MCA; (2) an abnormal vascular network near stenotic occlusion lesions in the arterial phase; and (3) bilateral presence of these manifestations [16]. Cerebral digital subtraction angiography or MR angiography were carried out for all MMD patients. Patients with any other disease that might explain the arterial steno-occlusive disease were excluded. The surgical treatment for MMD was based mainly on criteria, including symptoms of ischemia or hemorrhage, as well as the condition warranting surgery of the patients. Patients were eligible for further analyses only when: (1) Patients were aged ≥ 18 and diagnosis with MMD was confirmed through neuroimaging studies. (2) Their medical records were complete, and they had undergone surgical revascularization. (3) Patients successfully received direct or combined revascularization. The different revascularization procedures were categorized as direct revascularization, which primarily involved STA-MCA bypass, and indirect bypass, referring to methods like dura-synangiosis, myo-synangiosis, angio-synangiosis, or encephalo-duro-arterio-synangiosis. Combined revascularization integrated these two methods. Direct or combined revascularization were typically preferred unless donor or recipient vessels were too small (<0.8 mm), patients were overly fragile, or patients were very young. In situations where no suitable donor artery was available, an indirect bypass, such as dura-synangiosis, myo-synangiosis, angio-synangiosis or encephalo-duro-arterio-synangiosis, was chosen.

### 2.4. Clinical Follow-Up

Based on EMR data from the statistical system of the Health Commission of Hubei Province, all individuals in this study were followed up until 31 December 2021, for a minimum of one year. From 1 January 2019 to 31 December 2020, 4078 MMD patients were admitted to hospitals in Hubei. Revascularization was performed on 679 of these patients. With those aged < 18 and those with insufficient medical records excluded, 559 were finally included in this study. The detailed flowchart of this study is presented in Figure 1. Based on postoperative cerebrovascular events, the subjects were divided into three groups: (1) patients with no postoperative stroke (NPS); (2) patients with postoperative hemorrhagic stroke (PHS); and (3) patients with postoperative ischemic stroke (PIS). PHS was defined as cerebral hemorrhaging and subarachnoid hemorrhaging that occurred one month after revascularization, whereas PIS was classified as cerebral infarction and transient ischemic attack after one month of the initial operation. In this study, the following demographic and clinical characteristics were included as potential risk factors for postoperative cerebrovascular events in MMD patients: age, gender, hypertension, hyperlipidemia, diabetes mellitus, atherosclerosis, intracranial aneurysm, preoperative stroke, surgical procedures, Suzuki stage, admission modified Rankin scale (mRS), and postoperative presentation. All patients who were followed up had at least one vascular examination, such as CTA, DSA or MRI.

### 2.5. Statistical Analysis

Statistical analysis was performed using SPSS (Version 26.0, IBM, Chicago, IL, USA). The Kolmogorov–Smirnov test is used to test for normality. Normally distributed continuous data were expressed as mean ± standard deviation (x ± sd), and the independent-sample t-test was used for analysis. The chi-square test or Fisher’s exact test were used to compare categorical variables reported as frequencies. Stroke-free survival analysis was performed using Kaplan–Meier curves. Multivariate logistic regression was performed to analyze multivariate data to identify risk factors for postoperative cerebrovascular events. *p* < 0.05 was considered statistically significant.

## 3. Results

### 3.1. General Patient Characteristics

In this study, 559 adult patients (97 indirect revascularization, 105 direct revascularization, and 357 combined revascularization) with at least one year of follow-up (with an average follow-up duration of 25.1 months) were identified from the MMD database (Table 1). The adult MMD patients consisted of 286 females (51.2%) and 273 males (48.8%) (Figure 2A). Among them, PHS (3.0%) was observed in 17 cases, and PIS (7.7%) was seen in 43 cases (Figure 2B), presenting average yearly incidences of 1.5% and 3.7%, respectively. According to the Kaplan–Meier analysis, the estimated mean stroke-free interval for the entire cohort was 34.8 months (Figure 3). 

The age distribution of these MMD patients was predominantly concentrated within the range from 46 to 55 years, with a mean age of 49.1 ± 10.0 years. As illustrated in Figure 2C, the rate of combined revascularization was higher than that of direct and indirect revascularization alone. The initial symptom was ischemia and hemorrhage in 356 and 203 patients, respectively. Additionally, ischemic symptoms presented a greater prevalence compared to hemorrhagic symptoms across all age groups (Figure 2D).

### 3.2. Comparison of Background Factors

The baselines of included potential risk factors and patients’ demographic information were compared in Table 2. Of hemorrhagic patients, 12 had PHS. Compared to the NPS group, the rate of PHS was significantly higher in the hemorrhagic MMD group (*p* = 0.003). Additionally, the proportion of patients with hypertension (*p* = 0.003) and diabetes mellitus (*p* = 0.003) was significantly higher in the PIS than that in NPS, and the admission mRS was higher in PIS (*p* = 0.0034). Among the cohort of patients who remained free from postoperative stroke events, the majority underwent combined surgery, while a comparable proportion underwent direct and indirect revascularization procedures (18.2% and 17.8%, respectively). Notably, within the subset of patients who experienced PHS, 47.1% underwent combined surgery. In contrast, within the group of patients who developed postoperative ischemic stroke (PIS), a significantly higher proportion of 69.8% (n = 30) underwent combined surgery. Remarkably, among the patients in the PIS group, those who underwent indirect revascularization exhibited the lowest proportion of cases. However, PHS/PIS events in three surgical procedure groups exhibited no statistical significance compared to NPS. Meanwhile, no significant differences were observed in the proportions of hypertension, hyperlipidemia, atherosclerosis, intracranial aneurysm and surgical procedures between PHS and NPS. 

### 3.3. Identification of Risk Factors of Postoperative Stroke for MMD

Direct revascularization involves the direct reconnection and anastomosis of arteries, while indirect bypass refers to gradually reconstructing blood flow over a period of 2–3 months through methods such as arterio-synangiosis, dura-synangiosis, and temporal muscle synangiosis. Combined revascularization is a combination of these two approaches. Therefore, when grouping, we place direct bypass and combined bypass in the same group because they both involve the short-term reconstruction and changes in blood flow. 

The logistic regression analyses are presented in Table 3 and Table 4. Notably, preoperative hemorrhagic stroke was significantly associated with PHS (odds ratio [OR], 4.229 [95% CI, 1.244–14.376]; *p* = 0.021), indicating the higher risk MMD faced by patients with the hemorrhagic onset of cerebrovascular events after revascularization. Contrastingly, hypertension (odds ratio [OR], 0.424 [95% CI, 0.210–0.855]; *p* = 0.017), diabetes mellitus (odds ratio [OR], 0.368 [95% CI, 0.163–0.827]; *p* = 0.016) and admission mRS (odds ratio [OR], 2.301 [95% CI, 1.157–4.575]; *p* = 0.017) were associated with an increased risk of PIS.

## 4. Discussion

The study enrolled 559 adult MMD patients who underwent surgical revascularization. A comprehensive exploration into several background factors related to this subset of patients was conducted. It was discerned that there was no significant variance in the gender distribution among adult MMD patients undergoing revascularization procedures. A distinct age pattern emerged, with a notable peak in the 46–55-year range. Ischemic symptoms were more prevalent than hemorrhagic symptoms across all age groups, offering insights into the symptomatic profile of adult MMD patients receiving surgical treatment. Additionally, a correlation was observed between preoperative hemorrhagic events and PHS, with preoperative hemorrhage identified as a risk factor for PHS. This finding highlights its critical influence on postoperative outcomes. Comparing patients with PIS to those without postoperative strokes (NPS), notable differences were observed. Patients in the PIS category had higher incidences of hypertension and diabetes, along with elevated admission mRS scores. These results indicate a greater vulnerability to PIS in MMD patients with certain clinical characteristics, providing important information for clinical prognosis and management.

Several clinical features of MMD patients differed from those reported in previous studies in China. In a Beijing epidemiological study [17], the peak age for adult MMD patients was found to be 36–40 years. However, the present study revealed a different pattern, with the peak age for adult MMD patients being 46–55 years, which was higher than that observed in the Beijing study. It was hereby hypothesized that regional factors might contribute to the observed difference in the peak age of MMD patients. Additionally, it should be noted that the study population specifically included patients that had undergone surgical revascularization. As such, the mean age of patients requiring surgical intervention might be higher than that of the overall MMD patient population.

Overall, the findings from this comparative study provide valuable insights into potential predictors for postoperative stroke in patients with MMD. While risk factors for postoperative stroke have been explored in multiple retrospective studies [12,13,18,19], the precise underlying cause of postoperative hemorrhage and ischemia remains elusive. In the current study, MMD patients with preoperative hemorrhage were found to be more likely to have PHS than patients from the control group. Several pathological features might underlie this hemorrhagic tendency. Notably, regular layering of the elastic lamina was observed in the tunica media [20], and an elevated level of apoptosis was detected in MMD vessels [21]. Both of these factors could cause the enlargement and dilation of pre-existing perforator branches [22], thereby facilitating the formation of micro-aneurysms. Although surgical revascularization effectively regulated blood flow in the impaired hemisphere, subsequent hyperfusion, vulnerable MMD vessels, and micro-aneurysms also had potential to occur, which posed potential risks for postoperative rebleeding. Furthermore, MMD patients presenting with hemorrhagic symptoms might have more fragile perforating arteries [19], thereby increasing the likelihood of postoperative hemorrhage following blood flow disruption [23].

A previous study found an association between hypertension [24] and the onset pattern of MMD, but the relation between hypertension and postoperative stroke remains unclear. Furthermore, perioperative blood pressure management for adult MMD patients is still controversial [25]. A recent retrospective study involving 122 MMD patients that had received direct surgical revascularization presented the idea that induced hypertension and hypervolemia may not be necessary for preventing PIS in hemorrhagic MMD patients [26]. In the present study, however, hypertension was found to be associated with PIS, which might be attributed to the larger sample size in this study, the involvement of all MMD patients in Hubei province rather than only hemorrhagic patients, as well as the inclusion of all surgical procedures. The studies available have also indicated that MMD patients may have a higher odds ratio of having diabetes mellitus [27], and that the stenosis of major intracranial arteries may have an autoimmune etiology in patients with diabetes [28]. Herein, the diagnosis of diabetes was also found to be significantly associated with PIS, and the admission mRS score was a valuable component of MMD assessments. The present study demonstrated that a higher admission mRS score served as a risk factor for PIS. This finding underscores the importance of vigilant monitoring for patients with elevated admission mRS scores.

Direct revascularization was related to postoperative hyperperfusion, which might result in hemorrhagic stroke [29]. Herein, out of the 559 MMD inpatients, 462 (82.6%) underwent direct or direct/combined revascularization, whereas 97 (17.4%) had indirect revascularization. Among 17 inpatients who suffered PHS, 70.6% received direct or direct/combined revascularization, but the difference was not statistically significant. Meanwhile, the postoperative stroke rate for direct/combined revascularization was comparable to that for indirect revascularization [12].

This retrospective study amassed an extensive dataset from adult MMD inpatients’ EMRs in Hubei Province. The patients were hospitalized for a long time, and so the researchers were able to fully capture the details of their postoperative outcomes. Due to its retrospective design, the study identified and outlined various background factors associated with postoperative cerebrovascular events and proposed several risk factors contributing to these occurrences. Given the severity and relative rarity of MMD, the findings of this study are significant, positioning it as a valuable resource for developing strategies in postoperative MMD management. As we navigate the landscape of this rare condition, insights gained not only augment the existing body of knowledge but also offer potential for shaping effective postoperative care methodologies for individuals affected by MMD.

## 5. Advantages and Limitations

The data used in this study were collected from hospitals throughout Hubei Province and were both credible and traceable. Given that postoperative stroke was largely treated in local hospitals, data concerning these events became more accessible and extensive via EMRs. This methodology facilitated the assembly of a more extensive dataset, thereby fostering a nuanced comprehension of postoperative stroke incidents. Furthermore, the availability of comprehensive data from diverse local hospitals holds significant potential for shaping healthcare policies. The wealth of information sourced from various entities within Hubei Province can play a crucial role in the formulation of targeted interventions and advancements in postoperative stroke management at a regional level.

However, this study was still subject to certain limitations. Although differences were observed in several potential risk factors, the sample size of the PHS and PIS groups was still relatively small, and the differences in cerebrovascular event rates between the three groups were most likely smaller than hypothesized during the study conception, failing to effectively demonstrate the statistical significance. This limitation underscores the importance of exercising caution when interpreting the findings, especially in instances with smaller sample sizes. Additionally, data pertaining to patients with Moyamoya disease (MMD) were sourced from various hospitals, each with its own distinct surgical experiences. This variability in surgical approaches across different institutions introduces a potential confounding factor, as the effectiveness of surgeries may vary based on the expertise and protocols of individual medical facilities.

## 6. Conclusions

Overall, this study will enhance people’s understanding of the clinical features and risk factors of postoperative stroke in adult MMD. To be specific, we found that the age distribution of adult MMD patients with revascularization was predominantly concentrated within the range from 46 to 55 years. Ischemic symptoms presented a greater prevalence compared to hemorrhagic symptoms. Preoperative hemorrhage was identified to be the risk factor for PHS. Furthermore, hypertension, diabetes and higher admission mRS were identified as the risk factors for PIS. These results indicated the possibility of enhancing systematic disease management in order to contribute to the prevention of postoperative cerebrovascular accidents.

## Figures and Tables

**Figure 1 brainsci-13-01696-f001:**
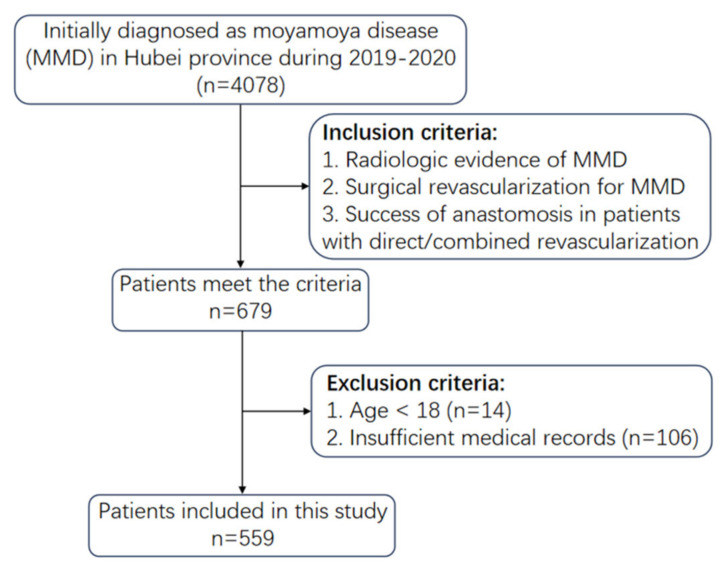
Flow diagram of filtration of study subjects.

**Figure 2 brainsci-13-01696-f002:**
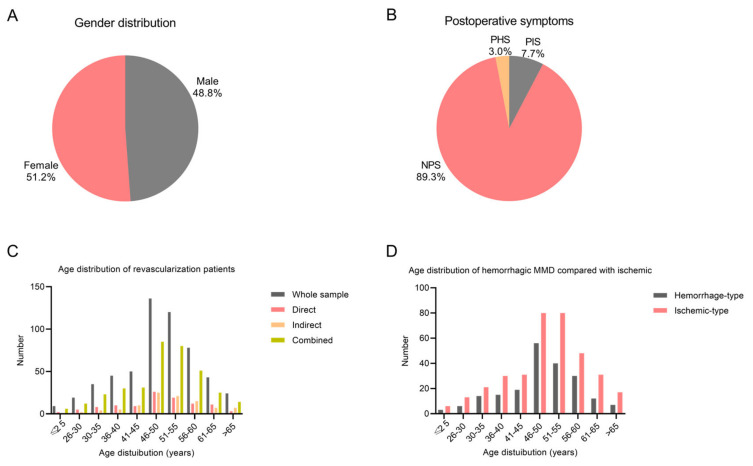
(**A**): Gender distribution of adult MMD patients with revascularization. (**B**): Postoperative symptoms distribution of adult MMD patients with revascularization. (**C**): Age distribution of adult MMD patients with direct, combined and indirect revascularization. (**D**): Age distribution of adult MMD patients with hemorrhagic MMD compared with ischemic stroke.

**Figure 3 brainsci-13-01696-f003:**
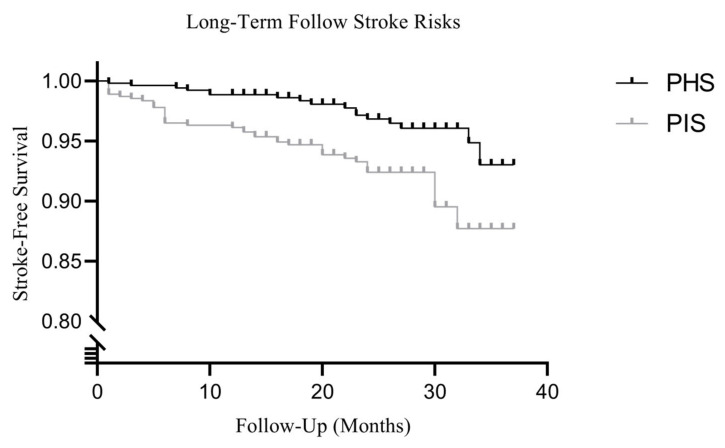
Kaplan–Meier cumulative hazard curve for long-term major hemorrhagic and ischemic risk in MMD patients post-revascularization.

**Table 1 brainsci-13-01696-t001:** Clinical Characteristics of the Study Population.

Characteristics	All Patients	Percentage (%)
Number of surgeries	n = 559	100
Age	49.1 ± 10.0	
Gender		
Female	286	51.2
Male	273	48.8
Hypertension		
Yes	220	39.4
NO	339	60.6
Hyperlipidemia		
Yes	31	5.5
NO	528	94.5
Diabetes Mellitus		
Yes	73	13.1
NO	486	86.9
Preoperative Presentation		
Hemorrhage	203	36.3
Ischemia	356	86.9
Atherosclerosis		
Yes	50	8.9
NO	509	91.1
Intracranial Aneurysm		
Yes	74	13.2
NO	485	86.8
Surgical Procedure		
Indirect	97	17.4
Direct	105	18.8
Combined	357	63.9
Suzuki Stage		
2	98	17.5
3	227	40.6
4	209	37.4
5	21	3.8
6	4	0.7
Admission mRS		
0–2	347	62.1
3–6	212	37.9
Postoperative Presentation		
Normal	499	89.3
Hemorrhage	17	3.0
ICH	9	1.6
IVH	3	0.5
SAH	1	0.2
ICH+IVH	4	0.7
Ischemia	43	7.7
TIA	17	3.0
Cerebral infarction	26	4.7

ICH, intracerebral hemorrhage; IVH, intraventricular hemorrhage; SAH, subarachnoid hemorrhage; TIA, transient ischemic attacks.

**Table 2 brainsci-13-01696-t002:** Comparison of background factors of different postoperative symptoms.

Characteristics	Postoperative Symptom				
	NPS (n = 499) n (%)	PHS (n = 17) n (%)	*p* Value (PHS vs. NPS)	PIS (n = 43) n (%)	*p* Value (PIS vs. NPS)
Age	48.8 ± 10.3	52.0 ± 6.5	0.197	51.3 ± 6.8	0.116
Gender					
Female	251 (50.8)	11 (64.7)	0.243	24 (55.8)	0.488
Male	248 (49.7)	6 (35.3)		19 (44.2)	
Hypertension					
Yes	186 (37.3)	9 (52.9)	0.413	26 (60.5)	0.003
NO	313 (62.)	8 (47.1)		17 (39.5)	
Hyperlipidemia					
Yes	27 (5.4)	1 (5.9)	0.933	3 (7.0)	0.667
NO	472 (94.6)	16 (94.1)		40 (93.0)	
Diabetes Mellitus					
Yes	60 (12.0)	1 (5.9)	0.441	12 (27.9)	0.003
NO	439 (88.0)	16 (94.1)		31 (72.1)	
Preoperative Presentation					
Hemorrhage	178 (35.7)	12 (70.6)	0.003	13 (30.2)	0.474
Ischemia	321 (64.3)	5 (49.4)		23 (69.8)	
Atherosclerosis					
Yes	47 (9.4)	1 (5.9)	0.622	2 (4.7)	0.296
NO	452 (90.6)	16 (94.1)		41 (95.3)	
Intracranial Aneurysm					
Yes	62 (12.4)	4 (23.5)	0.178	8 (18.6)	0.246
NO	437 (87.6)	13 (76.5)		35 (81.4)	
Surgical Procedure					
Direct	91 (18.2)	4 (23.5)	0.333	10 (23.3)	0.174
Indirect	89 (17.8)	5 (29.4)		3 (7.0)	
Combined	319 (63.9)	8 (47.1)		30 (69.8)	
Suzuki Stage					
2	90 (18.0)	1 (5.9)	0.256	7 (16.3)	0.400
3	194 (38.9)	11 (64.7)		22 (51.2)	
4	190 (38.1)	5 (29.4)		14 (32.6)	
5	21 (4.2)	0 (0.0)		0 (0.0)	
6	4 (0.8)	0 (0.0)		0 (0.0)	
Admission mRS					
0–2	314 (62.9)	12 (70.6)	0.519	20 (46.5)	0.034
3–6	185 (37.1)	5 (29.4)		23 (53.5)	

**Table 3 brainsci-13-01696-t003:** Logistic regression of factors for PHS (OR: odds ratio; CI: confidence interval).

Variable	Coeffect	Standard Error	OR	95% CI	*p* Value
Age	0.029	0.03	1.029	0.971–1.091	0.336
Gender	−0.399	0.532	0.671	0.236–1.903	0.453
Hypertension	0.35	0.538	1.419	0.494–4.075	0.516
Hyperlipidemia	0.638	1.109	1.893	0.215–16.637	0.565
Diabetes mellitus	−0.502	1.104	0.606	0.07–5.27	0.65
Preoperative stroke (Hemorrhage)	1.442	0.624	4.229	1.244–14.376	0.021
Atherosclerosis	−0.283	1.078	0.754	0.091–6.228	0.793
intracranial aneurysm	0.31	0.631	1.363	0.396–4.695	0.624
Surgical procedure	0.075	0.6	1.077	0.333–3.491	0.901
Suzuki stage	−0.05	0.309	0.952	0.519–1.744	0.872
Admission mRS	−0.105	0.581	0.9	0.288–2.813	0.857

**Table 4 brainsci-13-01696-t004:** Logistic regression of factors for PIS (OR: odds ratio; CI: confidence interval).

Variable	Coeffect	Standard Error	OR	95% CI	*p* Value
Age	−0.011	0.019	0.989	0.953–1.027	0.579
Gender	0.142	0.338	1.152	0.594–2.234	0.675
Hypertension	−0.857	0.358	0.424	0.210–0.855	0.017
Hyperlipidemia	0.072	0.673	1.074	0.287–4.018	0.915
Diabetes mellitus	−1.001	0.414	0.368	0.163–0.827	0.016
Preoperative stroke (Hemorrhage)	−0.365	0.398	0.694	0.318–1.513	0.359
Atherosclerosis	1.241	0.776	3.459	0.756–15.824	0.11
Intracranial aneurysm	−0.47	0.455	0.625	0.256–1.524	0.301
Surgical procedure	−1.156	0.649	0.315	0.088–1.123	0.075
Suzuki stage	0.324	0.208	1.382	0.919–2.080	0.12
Admission mRS	0.833	0.351	2.301	1.157–4.575	0.017

## Data Availability

The data presented in this study are available on request from the corresponding author. The data are not publicly available due to privacy and ethical restrictions.

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
