# Peer review of "Clinical Features and Risk Factors of Postoperative Stroke in Adult Moyamoya Disease"

_brainsci, 2023, doi:10.3390/brainsci13121696_

Round 1
Reviewer 1 Report
Comments and Suggestions for Authors
I have some comments to the authors:
1. In the aim of the study please provide the nature of this study - prospective or retrospective.
2. the first sub-section of the material and methods section should be ethics.
3. please add information about the statistical test used to analyze the normal data distribution.
4. why did authors analyze data only from one year?
5. tables and figures have to be self-explanatory.
6. advantages and limitations should be better described and future perspectives.
7. Please provide the reference/references for each statement in the introduction.
Author Response
We sincerely thank the editor and all reviewers for their valuable feedback that we have used to improve the quality of our manuscript. The reviewer comments are laid out below in boldfaced character and specific concerns have been numbered. Our response is given in normal font and changes/additions to the manuscript are given in the yellow text.
Response to Reviewer 1
- 1. In the aim of the study please provide the nature of this study - prospective or retrospective.
A:Thank you for your feedback. We appreciate your valuable comments. In response to your suggestion, we have revised the Methods section of the abstract (line 19) to incorporate the term "retrospectively" in order to accurately convey the nature of our study. The revised sentence now reads as follows: "Data of 4,078 MMD inpatients were retrospectively collected across the whole secondary and higher-level hospitals of Hubei Province from January 2019 to December 2020." We believe this modification better reflects the retrospective design of our study.
- the first sub-section of the material and methods section should be ethics.
A: Thank you for your suggestion. We have moved the ethics part to the first sub-section of Methods section, which is in line 70-75.
- please add information about the statistical test used to analyze the normal data distribution.
A: Thanks for your valuable comments. We have revised the “Statistical analysis” as fifth sub-section of Methods, and provided more detailed description of the statistical test we used to analyze the normal data distribution. In line 125-128, changing the previous sentences to “The Kolmogorov-Smirnov test is used to test for normality. Normally distributed continuous data were expressed as mean±standard deviation (x±sd), and the independent-samples t-test was used for analysis.”
- why did authors analyze data only from one year?
A: We greatly appreciate your comment. The analysis was focused on data from the year 2019 onward due to the initiation of standardized electronic medical record reporting in Hubei Province starting from 2016. The data from the initial years were found to be non-standardized with limited volume and substantial missing information, which could potentially compromise the robustness and reliability of the analysis. Therefore, to ensure the quality and comprehensiveness of the dataset, we chose to concentrate our analysis on the more reliable and complete data from 2019 onward. And the data was collected from Jan. 2019 to Dec 2020, which contained 2 years of patients. Then we followed up until Dec.2021.
- tables and figures have to be self-explanatory.
A: We greatly appreciate your valuable feedback. Taking it into consideration, we have made the following improvements to address your suggestions:
We have emphasized the p-value column in Table 3 and Table 4 by bolding it. This modification aims to enhance the readability and emphasize the statistical significance of the results.
In response to your comment, we have revised Figure 1 to present a clearer workflow of our data processing. This modification allows for a more comprehensive understanding of how the data was collected and filtered for further statistical analysis.
Allow us to provide a brief explanation of each table and figure:
Figure 1: This flow diagram illustrates the step-by-step process of data collection and patient selection for our study. It provides a visual representation of the methodology employed in our research.
Table 1: This table presents the baseline clinical characteristics of the study population. It provides an overview of key demographic and clinical variables that were assessed at the beginning of the study.
Figure 2: This figure displays several important aspects of adult MMD patients with revascularization. Subfigure A showcases the gender distribution of these patients, while subfigure B illustrates the distribution of postoperative symptoms. Subfigure C compares the age distribution between adult MMD patients with direct, combined and indirect revascularization. Lastly, subfigure D compares the age distribution between adult MMD patients with hemorrhagic MMD and those with ischemic MMD.
Figure 3: This figure presents the Kaplan-Meier cumulative hazard curve, providing insights into the long-term major hemorrhagic and ischemic risk in MMD patients post-revascularization.
Table 2: This table presents the result of correlation analysis of background factors of different postoperative symptoms.
Table 3: This table presents the results of logistic regression analysis, specifically focusing on factors associated with postoperative hemorrhagic stroke.
Table 4: This table depicts the results of logistic regression analysis, highlighting factors associated with postoperative ischemic stroke.
We believe that these modifications and explanations will further enhance the clarity and interpretability of our findings. We sincerely appreciate your valuable input, which has undoubtedly contributed to the overall quality of our study.
- advantages and limitations should be better described and future perspectives.
A: Thank you very much for your suggestion. We have illustrated the advantages more in detail, and added limitations in Section 6 “Advantages and Limitations”. For future perspectives, we described the potential value for policy making and enhancing people’s understanding of the clinical features and risk factors of postoperative stroke in adult MMD.
In line 276-297, the Section 6 was changed to “The data used in this study were collected from hospitals throughout Hubei Province and were both credible and traceable. Given that postoperative stroke was largely treated in local hospitals, data concerning these events became more accessible and extensive by EMRs. This methodology facilitated the assembly of a more extensive dataset, thereby fostering a nuanced comprehension of postoperative stroke incidents. Furthermore, the availability of comprehensive data from diverse local hospitals holds significant potential for shaping healthcare policies. The wealth of information sourced from various entities within Hubei Province can play a crucial role in the formulation of targeted interventions and advancements in postoperative stroke management at a regional level.
However, this study was still subject to certain limitations. Though differences in several potential risk factors were observed, the sample size of the PHS and PIS groups was still relatively small, and the differences in cerebrovascular event rates between the three groups were most likely smaller than hypothesized during the study conception, failing to effectively demonstrate the statistical significance. This limitation underscores the importance of exercising caution when interpreting the findings, especially in instances with smaller sample sizes. Additionally, data pertaining to patients with Moyamoya disease (MMD) were sourced from various hospitals, each with its distinct surgical experiences. This variability in surgical approaches across different institutions introduces a potential confounding factor, as the effectiveness of surgeries may vary based on the expertise and protocols of individual medical facilities”.
- Please provide the reference/references for each statement in the introduction.
A: Thank you for your careful review. We apologize for any confusion caused by the misleading position of the reference. We have taken your feedback into consideration and made the necessary adjustment in line 60 (reference [7]) to ensure clarity. We deeply regret any inconvenience this may have caused and appreciate your understanding.
Reviewer 2 Report
Comments and Suggestions for Authors
This is an article from Hubei China describing a retrospective cohort of patients from January 2019- December 2020 (2 years). A total of 4,078 patients were identified across all hospitals in Hubei province during this period. After inclusion and exclusion criteria were applied a total of 559 adult MMD patients were included in this study. The median follow-up time was 25 months. Most of the patients (83%) underwent combined/direct revascularization. There were a total of 17 (3%) post operative hemorrhagic strokes and 43 post operative ischemic strokes (8%). They conclude that patients presenting with preoperative hemorrhage had a higher risk for postoperative hemorrhage and that diabetes, hypertension, and a poor mRS correlated with post operative stroke.
Specific Comments:
· The authors state that there are few studies that factors for stroke in adults with MMD (2021). There was a recent large meta-analysis published in Stroke that evaluated post operative hemorrhage and ischemic events.1
· There is very little information about the surgical procedures. Were these all performed by a small group of surgeons with experience? Who did these surgeries? If the surgeries are done differently there is potential for bias.
· What was the status of the bypass at the time of hemorrhage and stroke? They should include Matsushima scales and patency rates. They should also list reasons for why they postulate patients had ischemia following surgeries – was it technical failure of the bypass, insufficient flow, low blood pressure. Were the strokes all localized to the same hemisphere?
· How many of these patients had Moyamoya syndrome instead of disease? They list 10% as having atherosclerosis.
· Were these major or minor strokes? They should consider dividing these up as that makes a clinical difference in outcome.
· They should divide the pt’s with indirect vs combined and compare how each cohort did.
Overall, this could be an interesting addition to the literature with the number of patients included. However, the material is not novel and needs better analysis and comparisons to be useful to the literature.
· 1. Nguyen VN, Motiwala M, Elarjani T, et al. Direct, Indirect, and Combined Extracranial-to-Intracranial Bypass for Adult Moyamoya Disease: An Updated Systematic Review and Meta-Analysis. Stroke. Sep 22 2022:101161STROKEAHA122039584. doi:10.1161/STROKEAHA.122.039584
Comments on the Quality of English Language
English is acceptable.
Author Response
We sincerely thank the editor and all reviewers for their valuable feedback that we have used to improve the quality of our manuscript. The reviewer comments are laid out below in boldfaced character and specific concerns have been numbered. Our response is given in normal font and changes/additions to the manuscript are given in the yellow text.
Response to Reviewer 2
Comment1: The authors state that there are few studies that factors for stroke in adults with MMD (2021). There was a recent large meta-analysis published in Stroke that evaluated post operative hemorrhage and ischemic events.
A: Thank you for your kind suggestion and we appreciate your deep insight into MMD research. The statement “Studies on the clinical features and risk factors for postoperative stroke after surgical revascularization in adult moyamoya disease (MMD) have been rarely reported” is not proper. And it was changed to “The clinical features and risk factors for postoperative stroke after surgical revascularization in adult moyamoya disease (MMD) have not been fully elucidated”.
We greatly appreciate your comment. We have carefully reviewed the article 'Direct, Indirect, and Combined Extracranial-to-Intracranial Bypass for Adult Moyamoya Disease: An Updated Systematic Review and Meta-Analysis.' While acknowledging its valuable focus on comparing different surgical groups, we would like to highlight that our study takes a different approach by analyzing our data based on the comparison of post-operative stroke events, which may form a different perspective. We sincerely thank you for bringing attention to this distinction.
Comment2: There is very little information about the surgical procedures. Were these all performed by a small group of surgeons with experience? Who did these surgeries? If the surgeries are done differently there is potential for bias.
A: Thank you for your comment regarding the surgical procedures in our study. Our data was obtained from a comprehensive provincial electronic medical record system, which includes records from multiple hospitals and various surgeons across thirteen cities. Unfortunately, due to the nature of the data source, we do not have access to specific details regarding individual surgeons or the variations in surgical procedures employed. It is important to acknowledge that this limitation may introduce potential bias into our study findings. And we have mentioned it as a source of bias in Section6 “Advantages and limitations”
Comment3: What was the status of the bypass at the time of hemorrhage and stroke? They should include Matsushima scales and patency rates. They should also list reasons for why they postulate patients had ischemia following surgeries – was it technical failure of the bypass, insufficient flow, low blood pressure. Were the strokes all localized to the same hemisphere?
A: Thank you for your inquiry regarding the status of bypass and specific details related to hemorrhage, stroke, and ischemia in our study. We would like to highlight that our data is derived from a comprehensive provincial electronic medical record system, which does not provide access to the specific procedural details you mentioned. Unfortunately, we are unable to provide information on parameters such as Matsushima scales, bypass patency rates, reasons for postulated ischemia, technical failure of the bypass, insufficient flow, low blood pressure, or the localization of strokes to specific hemispheres. These details are beyond the scope of our available data. We appreciate your understanding of the limitations imposed by the nature of our data source.
Comment4: How many of these patients had Moyamoya syndrome instead of disease? They list 10% as having atherosclerosis.
A: Thank you for your question. The diagnosis, which was based on clinically accepted guidelines1, was made by different doctors in different hospitals across Hubei Province. And we have excludbased on clinically accepted guidelinesed patients diagnosed with Moyamoya syndrome. We are terribly sorry for the missed information. And we have modified our Flow chart in Figure1, which now gives more detailed exclusion criteria.
1.Guidelines for diagnosis and treatment of moyamoya disease (spontaneous occlusion of the circle of Willis). Neurol Med Chir (Tokyo), 2012. 52(5): p. 245-66.
Comment5: Were these major or minor strokes? They should consider dividing these up as that makes a clinical difference in outcome.
A: Thank you for emphasizing the importance of including stroke severity in our study. We completely agree with you on its relevance. However, we are sorry to inform you that our data source, which relies on hospital reports from Hubei Province, only provides information on the type of stroke, and not the severity. Unfortunately, we do not have access to specific details regarding the severity of strokes in our dataset. We acknowledge that this limitation restricts our ability to explore the impact of stroke severity on the outcomes studied. We appreciate your understanding of the constraints imposed by our data source and value your feedback for future research considerations.
Comment6: They should divide the pt’s with indirect vs combined and compare how each cohort did.
A: Thank you for your suggestions. We have divided the patients with three major forms of revascularization (direct, combined and indirect). In table 1, we found that the 559 patients who met the inclusion criteria consisted of 97 indirect , 105 direct , and 357 combined revascularization patients. However, there was no statistical correlation between postoperative symptom and the mode of revascularization (table 2).
Direct revascularization involves the direct reconnection and anastomosis of arteries, while indirect bypass refers to gradually reconstructing blood flow over a period of 2-3 months through methods such as arterio-synangiosis, dura-synangiosis, and temporal muscle -synangiosis. Combined revascularization is a combination of these two approaches. Therefore, when grouping, we place direct bypass and combined bypass in the same group because they both involve the short-term reconstruction and changes in blood flow. We have added the rationale for grouping in Section 3.3, and thank you for you insight and suggestion for us.
Table 1. Clinical Characteristics of the Study Population.
|
Characteristics |
All Patients |
Percentage (%) |
|
Number of surgeries |
n=559 |
100 |
|
Age |
49.1±10.0 |
|
|
Gender |
||
|
Female |
286 |
51.2 |
|
Male |
273 |
48.8 |
|
Hypertension |
||
|
Yes |
220 |
39.4 |
|
NO |
339 |
60.6 |
|
Hyperlipidemia |
||
|
Yes |
31 |
5.5 |
|
NO |
528 |
94.5 |
|
Diabetes Mellitus |
||
|
Yes |
73 |
13.1 |
|
NO |
486 |
86.9 |
|
Preoperative Presentation |
||
|
Hemorrhage |
203 |
36.3 |
|
Ischemia |
356 |
86.9 |
|
Atherosclerosis |
|
|
|
Yes |
50 |
8.9 |
|
NO |
509 |
91.1 |
|
Intracranial Aneurysm |
||
|
Yes |
74 |
13.2 |
|
NO |
485 |
86.8 |
|
Surgical Procedure |
||
|
Indirect |
97 |
17.4 |
|
Direct |
105 |
18.8 |
|
Combined |
357 |
63.9 |
|
Suzuki stage |
||
|
2 |
98 |
17.5 |
|
3 |
227 |
40.6 |
|
4 |
209 |
37.4 |
|
5 |
21 |
3.8 |
|
6 |
4 |
0.7 |
|
Admission mRS |
||
|
0-2 |
347 |
62.1 |
|
3-6 |
212 |
37.9 |
|
Postoperative Presentation |
||
|
Normal |
499 |
89.3 |
|
Hemorrhage |
17 |
3.0 |
|
ICH |
9 |
1.6 |
|
IVH |
3 |
0.5 |
|
SAH |
1 |
0.2 |
|
ICH+IVH |
4 |
0.7 |
|
Ischemia |
43 |
7.7 |
|
TIA |
17 |
3.0 |
|
Cerebral infarction |
26 |
4.7 |
ICH, intracerebral hemorrhage; IVH, intraventricular hemorrhage; SAH, subarachnoid hemorrhage; TIA, transient ischemic attacks.
Table 2. Comparison of background factors of different postoperative symptoms
|
Characteristics |
Postoperative Symptom |
||||
|
NPS (n=499) n(%) |
PHS (n=17) n(%) |
p Value (PHS vs. NPS) |
PIS (n=43) n(%) |
p Value (PIS vs. NPS) |
|
|
Age |
48.8±10.3 |
52.0±6.5 |
0.197 |
51.3±6.8 |
0.116 |
|
Gender |
|||||
|
Female |
251 (50.8) |
11 (64.7) |
0.243 |
24 (55.8) |
0.488 |
|
Male |
248 (49.7) |
6 (35.3) |
19 (44.2) |
||
|
Hypertension |
|||||
|
Yes |
186 (37.3) |
9 (52.9) |
0.413 |
26 (60.5) |
0.003 |
|
NO |
313 (62.) |
8 (47.1) |
17 (39.5) |
||
|
Hyperlipidemia |
|||||
|
Yes |
27 (5.4) |
1 (5.9) |
0.933 |
3 (7.0) |
0.667 |
|
NO |
472 (94.6) |
16 (94.1) |
40 (93.0) |
||
|
Diabetes Mellitus |
|||||
|
Yes |
60 (12.0) |
1 (5.9) |
0.441 |
12 (27.9) |
0.003 |
|
NO |
439 (88.0) |
16 (94.1) |
31 (72.1) |
||
|
Preoperative Presentation |
|||||
|
Hemorrhage |
178 (35.7) |
12 (70.6) |
0.003 |
13 (30.2) |
0.474 |
|
Ischemia |
321 (64.3) |
5 (49.4) |
23 (69.8) |
||
|
Atherosclerosis |
|||||
|
Yes |
47 (9.4) |
1 (5.9) |
0.622 |
2 (4.7) |
0.296 |
|
NO |
452 (90.6) |
16 (94.1) |
41 (95.3) |
||
|
Intracranial Aneurysm |
|||||
|
Yes |
62 (12.4) |
4 (23.5) |
0.178 |
8 (18.6) |
0.246 |
|
NO |
437 (87.6) |
13 (76.5) |
35 (81.4) |
||
|
Surgical Procedure |
|||||
|
Direct |
91 (18.2) |
4 (23.5) |
0.333 |
10 (23.3) |
0.174 |
|
Indirect |
89 (17.8) |
5 (29.4) |
3 (7.0) |
||
|
Combined |
319 (63.9) |
8 (47.1) |
30 (69.8) |
||
|
Suzuki stage |
|||||
|
2 |
90 (18.0) |
1 (5.9) |
0.256 |
7 (16.3) |
0.400 |
|
3 |
194 (38.9) |
11 (64.7) |
22 (51.2) |
||
|
4 |
190 (38.1) |
5 (29.4) |
14 (32.6) |
||
|
5 |
21 (4.2) |
0 (0.0) |
0 (0.0) |
||
|
6 |
4 (0.8) |
0 (0.0) |
0 (0.0) |
||
|
Admission mRS |
|||||
|
0-2 |
314 (62.9) |
12 (70.6) |
0.519 |
20 (46.5) |
0.034 |
|
3-6 |
185 (37.1) |
5 (29.4) |
23 (53.5) |
||
Reviewer 3 Report
Comments and Suggestions for Authors
The manuscript is interesting and has practical value, but needs major correction.
1. In materials and methods: What methods were used for follow-up monitoring (angiography, MRI, cerebral blood flow studies)?
2. Page 4 line 127: What do you mean by direct/combined revascularization? If this mean a combination of direct revascularization and the use of donor tissue, then “combined revascularization” should be written; or by this mean two groups of patients, some of whom underwent direct revascularization and some who underwent combined revascularization?
3. In indirect revascularization, were the same donor tissues used in all cases?
4. Which criteria were used to select patients for direct/combined or indirect revascularization?
5. In the section 3.2 (Page 6) the authors note that in 2.8% of the patients with direct/combined revascularization was revealed PHS and in 8.4 % - PIS. In the group of patients with indirect revascularization PHS was detected in 5.2% and PIS in 3.1%. In addition to pre-operatives risk factors and clinical features, is such a difference between the percentages related to the surgical tactics (especially in relation to PHS)?
6. Were there cases of anastomosis failure in patients with direct/combined revascularization? If there were no anastomosis failure, this should also be noted in the results.
7. Were there any intraoperative complications? If there were no intraoperative complications, this should also be noted in the results.
Author Response
We sincerely thank the editor and all reviewers for their valuable feedback that we have used to improve the quality of our manuscript. The reviewer comments are laid out below in boldfaced character and specific concerns have been numbered. Our response is given in normal font and changes/additions to the manuscript are given in the yellow text.
Response to Reviewer3:
- 1. In materials and methods: What methods were used for follow-up monitoring (angiography, MRI, cerebral blood flow studies)?
A: Thank you for your comment regarding the follow-up monitoring. All the patients were followed up with at least one type of cerebrovascular imaging (CTA DSA or MRI), and we have added this part to Section2.4 “Clinical Follow-up”. We are grateful for your understanding of the constraints imposed by our data source and value your feedback for future research considerations.
- 2. Page 4 line 127: What do you mean by direct/combined revascularization? If this mean a combination of direct revascularization and the use of donor tissue, then “combined revascularization” should be written; or by this mean two groups of patients, some of whom underwent direct revascularization and some who underwent combined revascularization?
A: Thank you for your response. We appreciate your clarification regarding the grouping of revascularization in our study. Allow us to provide a more detailed description:
The group labeled "direct/combined revascularization" consisted of patients who underwent direct revascularization procedures, either as standalone bypass or in combination with indirect revascularization techniques.
On the other hand, the group referred to as "indirect revascularization" comprised patients who exclusively underwent indirect revascularization procedures without any direct revascularization techniques.
In the revised manuscript, we have grouped our patients into 3 groups, direct, combined and indirect revascularization in Table2 to make the grouping more reasonable.
- In indirect revascularization, were the same donor tissues used in all cases?
A: Thank you for your question. In our study, the category of indirect revascularization encompassed dura-synangiosis, myo-synangiosis, angio-synangiosis or encephalo-duro-arterio-synangiosis. Different indirect revascularization procedures were selected according to the different conditions of patients. We hope this explanation clarifies the nature of the procedures included in the indirect revascularization group in our study.
- 4. Which criteria were used to select patients for direct/combined or indirect revascularization?
A: We appreciate your question. Direct/combined revascularization was typically the preferred treatment, unless the donor/recipient vessels were deemed too small (< 0.8 mm), excessively fragile, or in cases involving very young patients. In situations where no suitable donor artery was available, an indirect bypass, such as dura-synangiosis, myo-synangiosis, angio-synangiosis or encephalo-duro-arterio-synangiosis, was chosen.
- 5. In the section 3.2 (Page 6) the authors note that in 2.8% of the patients with direct/combined revascularization was revealed PHS and in 8.4 % - PIS. In the group of patients with indirect revascularization PHS was detected in 5.2% and PIS in 3.1%. In addition to pre-operative risk factors and clinical features, is such a difference between the percentages related to the surgical tactics (especially in relation to PHS)?
A: Thank you for your question. According to the results of logistic regression analysis, the p-value for different surgical procedure was 0.075 (PIS) and 0.901 (PHS), which did not present statistical significance. This may be due to the limitation of sample size, which needs further follow-up.
- Were there cases of anastomosis failure in patients with direct/combined revascularization? If there were no anastomosis failure, this should also be noted in the results.
A: Thank you for your suggestion. And we have excluded the cases with anastomosis failure. We have revised the flow chart in Figure1, to present our exclusion criteria more detailed.
Figure 1. Flow diagram of filtration of study subjects.
- Were there any intraoperative complications? If there were no intraoperative complications, this should also be noted in the results.
A: We sincerely appreciate your question. It is important to acknowledge that, given the nature of our data, we were only able to access the names of the surgical procedures performed, and unfortunately, detailed treatment information during surgery was not accessible to us. Regrettably, we were unable to obtain this specific information for our study. We apologize for the limitation in our ability to provide detailed treatment information during surgery. Your understanding of these constraints is highly valued.
Round 2
Reviewer 1 Report
Comments and Suggestions for Authors
None
Reviewer 2 Report
Comments and Suggestions for Authors
The authors have satisfactorily answered the queries. There are significant limitations that exist in this study that the study design will not be able to overcome (which they have acknowledged).
Comments on the Quality of English Language
English Language is acceptable.
Reviewer 3 Report
Comments and Suggestions for Authors
The authors took into account all my comments and answered all my questions thoroughly. I have no more questions.